# Restage4D: Reanimating Deformable 3D Reconstruction from a Single Video

Jixuan He[1] , Chieh Hubert Lin[2], Lu Qi[3],*Ming-Hsuan Yang[2]
[1]Cornell Tech, [2] University of California, Merced, [3] Wuhan University
jh2926@cornell.edu

## Abstract

Creating deformable 3D content has gained increasing attention with the rise of text-to-image and image-to-video generative models. While these models provide rich semantic priors for appearance, they struggle to capture the physical realism and motion dynamics needed for authentic 4D scene synthesis. In contrast, real-world videos can provide physically grounded geometry and articulation cues that are difficult to hallucinate. One question is raised: *Can we generate physically consistent 4D content by leveraging the motion priors of the real-world video*? In this work, we explore the task of reanimating deformable 3D scenes from a single video, using the original sequence as a supervisory signal to correct artifacts from synthetic motion. We introduce **Restage4D**, a geometry-preserving pipeline for video-conditioned 4D restaging. Our approach uses a video-rewinding training strategy to temporally bridge a real base video and a synthetic driving video via a shared motion representation. We further incorporate an occlusion-aware rigidity loss and a disocclusion backtracing mechanism to improve structural and geometry consistency under challenging motion. We validate Restage4D on DAVIS and PointOdyssey, demonstrating improved geometry consistency, motion quality, and 3D tracking performance. Our method not only preserves deformable structure under novel motion, but also automatically corrects errors introduced by generative models, revealing the potential of video prior in 4D restaging task. Source code and trained models will be released.

## 1 Introduction

Recent advances in foundational generative models in images [1, 2], videos [3], and 3D scenes [4, 5] drive growing interest in the creation of 4D content, which brings time-varying dynamics to 3D. In practical applications, controllability remains a valuable and continuing challenge in all generative frameworks. Inspired by video generative models, early controllable 4D generative frameworks condition on text prompts [6] or an image [7]. This intrigues us to explore other conditioning sources that are beneficial and intuitive for 4D content creation. In particular, we are interested in video conditioning, which is easily accessible on the Internet, exhibits physical properties of objects, and maintains visible regions that are broader than a single image. Meanwhile, existing works in 4D synthesis [8] mostly source the dynamics from video foundation models. Despite these models being able to synthesize visually appealing results with seemingly correct dynamics, they struggle to ensure physical plausibility, such as inconsistent appearance (*e.g.*, geometry modified after occlusion or out-of-view) and infeasible deformation (*e.g.*, limbs swapping during intersection). In Section 4.3, we show that extracting explicit 3D deformation from infeasible physics leads to floaters and broken geometry. These observations motivate us to develop a new paradigm for creating 4D content,

---

*Corresponding author: Lu Qi

39th Conference on Neural Information Processing Systems (NeurIPS 2025).

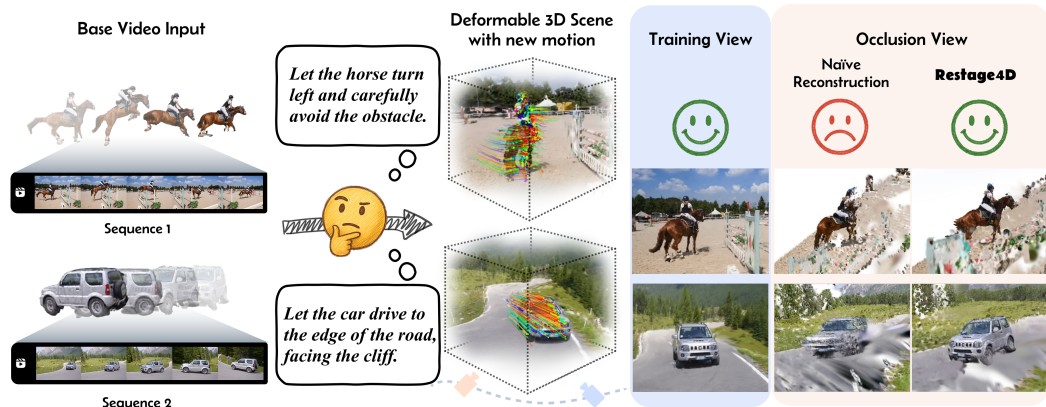

Figure 1: We show the input and output for the Restage4D in the left. Given a base video and a text prompt, we reanimate the scene according to the specified motion, while keeping the geometry consistency for occlusion part and motion coherent for the restaged 4D scene as shown in the right.

leveraging two promising information sources: the geometry and physical constraints distilled from a real-world video and the diverse pseudo-motion synthesized by video foundation models.

One intuitive way to build a 4D representation is using single-video deformable 3D reconstruction [9, 10], which achieves photorealistic results with temporally coherent and physically plausible deformations. Such methods commonly model dense 3D deformation using low-rank motion bases, then linearly combine via motion coefficients. These motion coefficients determine how the bases are weighted to produce the final deformation. Their similarity can be interpreted as an indicator of how tightly coupled the two spatial points are in the object's motion. We observe that such a design is conceptually similar to the skinning weights of articulated meshes [11]. In graphics workflow, the artists assign deformations to only a few key points, and per-vertex skinning weights map those deformations to the full mesh and drive the articulated meshes to perform desired motions. In this view, the input base video provides informed skinning weights in the real world, effectively turning the reconstructed object into a *puppet*, whose strings can be manipulated by altering the low-rank motion components. This intrigues us to explore controlling the puppet with dynamics generated by a video diffusion model.

We propose **Restage4D** towards a new *4D restaging* task to create video-conditioned 4D content. Given a monocular video, Restage4D aims to synthesize a new 4D motion sequence that maintains the original appearance while generating novel and physically plausible deformations. The restaged 4D sequence is driven by a synthetic video generated by an image-conditioned video diffusion model based on the first frame of the base video. More specifically, the base video provides the shape and articulation model (*i.e.*, the puppet to control), while the driving video supplies new dynamics (*i.e.*, manipulate the puppet) and the appearance of the additional dis-occlusion revealed by novel motions.

The first challenge lies in sharing a consistent shape and motion representation across both videos. We observe that the deformable 3D reconstruction frameworks are invariant to temporal direction, reconstructing from a video playing in a reversed temporal order (*i.e.*, *rewind*) would result in the same reconstruction. In combination with the assumption that our base video and driving video share the first frame, we propose a video-rewinding joint-training scheme. We rewind the base video into the reversed temporal order, temporally concatenate with the driving video, and then jointly reconstruct both sequences as a single video. This enables smooth motion transition across videos when initializing deformable 3D reconstruction frameworks with smooth tracking, and allows intuitive sharing of motion coefficients across two videos. However, the simple joint optimization does not sufficiently leverage all the available information. As the synthetic video presents novel motions, certain geometries visible in the input video become partially or even completely invisible throughout the synthetic video. Lacking visibility makes these geometries unable to receive gradients to preserve the smoothness and continuity of the appearance. To leverage the geometry recovered in the input video, we introduce an occlusion-aware rigidity regularization to preserve the local rigidity of the less visible geometry. In addition, we use a disocclusion backtracing mechanism to recover missing canonical geometry by tracing observed points from the driving video back to the base video.

We validate our method on two datasets with monocular video settings and complicated motions, DAVIS [12] and PointOdyssey [13], as well as several self-collected Internet videos to test the generalization. Our experimental results show that our Restage4D can correct both infeasible motion and inconsistent geometries created by video diffusion models under complex deformations. Under rigorous ablation and physically-based metrics evaluation, we show that our method achieves consistent improvements across reconstruction quality, appearance consistency, and physical consistency.

The main contributions of this work are:

- We introduce a new task, *4D restaging*, which aims to reanimate deformable 4D scenes from video-driven motion.
- We propose a geometry-preserving pipeline that leverages a video-rewinding joint training scheme, combined with occlusion-aware ARAP and disocclusion backtracing.
- We demonstrate how real-world supervision can correct artifacts from generative videos, bridging controllability and geometric fidelity in 4D reconstruction.

## 2 Related Work

**Deformable 3D Neural Rendering.** Neural rendering techniques based on NeRF [14, 15, 16, 17, 18, 19, 20], and Gaussian Splatting [21, 22] have demonstrated impressive performance in reconstructing high-fidelity static 3D scenes. Building upon this success, recent works have explored how to extend these representations to handle deformable 3D scenes with complex motion. A prominent line of work, including D-NeRF [23], Nerfies [24], and HyperNeRF [25], learns a time-conditioned deformation field via MLPs, enabling per-frame warping from a canonical space. However, because of the implicit nature of NeRF, maintaining coherent motion and preserving geometry over time remains challenging. To address this, recent Gaussian Splatting-based approaches [26, 27, 28, 29] adopt similar deformation-field formulations to model temporal variation, using multi-view videos as input. In contrast, methods like Shape-of-Motion [9], Mosca [10] and DGS-LRM [30] explicitly represent motion via rigid-body transformations and shared motion bases by utilizing depth [31], tracking [32] and camera-pos [33] priors. Although occlusion remains a significant challenge for such approaches, the explicit nature of the representation enables finer-grained control over motion and facilitates geometric reasoning. However, existing monocular approaches suffer from preserving geometric information under occlusion introduced by the complex motion. Therefore, our work focuses on maintaining geometric realism and motion coherence during monocular video-conditioned 4D content creation, which requires strong regularization and guidance from real observations.

**Motion Retargetting.** Motion retargeting is a fundamental problem in human-centric motion analysis and plays a central role in motion capture (MoCap) [34], scene animation [35], and motion transfer [36]. The goal of retargeting is to animate the static object using a reference input. Early works formulate retargeting as an optimization problem with kinematic constraints on articulationd body models [37], often solved via inverse kinematics (IK) [38, 39]. Recent approaches leverage deep learning by incorporating skeleton-aware modules [40, 41], enabling high-quality real-time motion retargetting in a variety of subjects. Beyond human motion, retargeting deformable 3D scenes presents new challenges. Under the Gaussian Splatting framework, methods such as SC-GS [42] and D-MiSo [43] introduce keypoint-based control to transfer motion from one 4D scene to another. Although effective in constrained scenarios, these methods rely on carefully designed control points and often require meticulous tuning to produce natural motion, especially when handling complex or long-range deformations. In contrast, our approach adopts an example-driven paradigm. Instead of designing explicit keypoint trajectories, we reanimate the scene by conditioning on a generated video sequence. This allows for retargeting complex and subtle motions directly from video while preserving geometric and temporal coherence.

**4D generation.** Recent advances in large-scale generative models such as Stable Diffusion [1], Flux [44], and Sora [45] have catalyzed significant progress in image, video and 3D generation. Naturally, this momentum has extended to 4D content generation, where the goal is to synthesize dynamic 3D scenes with time-varying geometry and appearance. One line of work directly uses generative models to create multiview observations or spatiotemporal sequences for the object condition on image or text , such as Zero-1-to-3 [46] and 4D-Diffusion [47]. Another line of research integrates synthetic supervision via score distillation sampling (SDS): methods like DreamGaussian [48] and

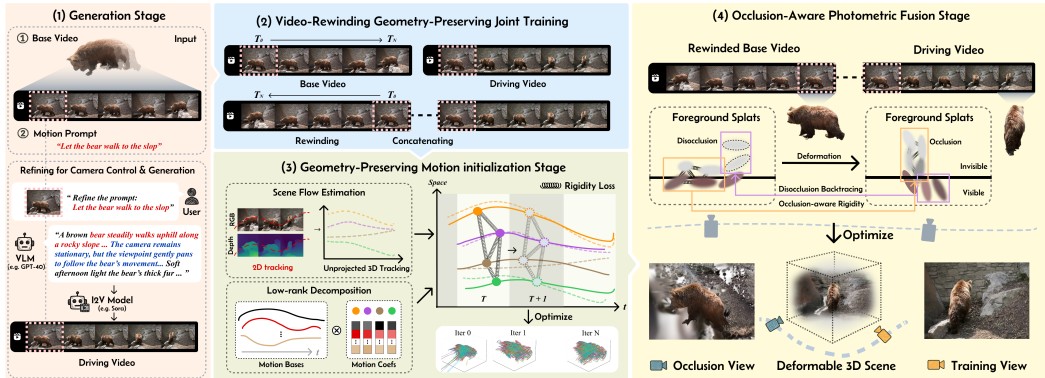

Figure 2: Pipeline of Restage4D. To perform a 4D restaging task, a base video and a text prompt are utilized to generate a driving video in the generation stage. Then, in video-rewinding stage, the base video is back-played and concatenate with the driving video to perform joint optimization. A geometry-preserving rigidity loss is applied in the motion initialization stage. Then, in photometric fusion stage, a occlusion-aware rigidity loss and a backtracing mechanism is incorporated.

4D-FY[8] use SDS to guide 3D or 4D reconstruction from text or reference images. Style-conditioned generation and editing also benefit from generative models, as demonstrated by works such as Instruct-NeRF2NeRF [49] and DreamBooth3D [50], where image-generated priors are used to control object appearance, geometry, or identity across views. However, few works explore using video as input for the objects' articulation and using text as motion source to generate realistic 4D scene. By using semantic prior in the video, our work tackles the problem of inconsistent appearance and motion in 4D restaging, helping to preserve physical plausibility and temporal coherence.

## 3 Method

With an input base video providing the context and a text prompt providing the description of motion, our goal is to create a 4D scene in which the object follows the specified motion. Since we use a monocular video as reference for the objects' motion articulation and appearance and a text prompt as the source of motion, it is very likely that the novel motion introduces occlusion and artifacts, leading to wrong geometry. To have a better geometry in the restaged 4D scene, the key is to utilize the prior knowledge in the base video. Therefore, we use a low-rank decomposed representation combined with a geometry-preserving training pipeline to obtain authentic 4D content.

### 3.1 Preliminaries: Low-rank Decomposed 4D Gaussian Splatting Representation

We adopt a motion representation similar to *Shape-of-Motion* [9] to model dynamic 4D Gaussian Splatting. Specifically, the scene is represented by two components: a static background point set $\mathbf{B} = \{b_0, b_1, ..., b_{N_b-1}\}$ and a dynamic foreground point set $\mathbf{X} = \{x_0, x_1, ..., x_{N_f-1}\}$, where each point is associated with a 3D Gaussian (mean, scale, rotation, opacity and color).

To capture temporal motion, each foreground point $x_i$ is associated with a canonical position $\mu_i \in \mathbb{R}^3$, a shared time-independent motion coefficient $\beta_i \in \mathbb{R}^K$, and a set of time-varying motion bases $\mathbf{M} = \{M_k(t)\}_{k=1}^K$, where each $M_k(t) \in \text{SE}(3)$ is a smooth rigid body transformation over time.

The deformation of each foreground point at time $t$ is obtained by composing the motion bases weighted by its coefficients:

$$T_i(t) = \gamma(\sum_{k=1}^{K} \beta_{ik} \cdot M_k(t)), \qquad x_i(t) = T_i(t) \cdot \mu_i \tag{1}$$

where the transformation is applied in SE(3), and $x_i(t)$ denotes the 3D position of point $x_i$ at time $t$. An orthogonalization function $\gamma$ is applied to ensure the legality of $T_i$. During training, the parameters of the Gaussians and the motion bases $M_k(t)$ are optimized to match multi-view input observations. The training procedure in Shape-of-Motion consists of two stages. In the motion initialization phase, a pre-trained tracking model is applied to detect dynamic foreground points across frames. These

tracked 2D points are then unprojected into 3D to estimate a coarse scene flow, providing supervision for the initial motion. Based on this, the motion coefficients $\beta_{ik}$ and the time-varying motion bases $M_k(t)$ are initialized and optimized to best fit the scene flow. In the photometric fusion phase, the model is fine-tuned using a standard Gaussian splatting-style inverse rendering pipeline, which jointly refines motion, appearance, and structure using photometric losses and 2D tracking consistency.

## 3.2 Driving Video Generation & Rewinding

To achieve 4D scene creation with novel motion, we propose an example-driven pipeline that incorporates the deformable object's geometry and motion articulation from the base video and the motion dynamics from the text prompt. As shown in the first stage of Figure 2, the input of the pipeline is a base video and a motion prompt describing the desired motion. To create the dynamics from the text prompt to drive the object, instead of using some key points to simulate the proposed motion, we leverage an Image-to-Video (I2V) diffusion model such as Sora [45] to synthesize a driving video for complex and plausible motion as the source of dynamics. To be specific, the first frame of the base video is extracted as the input for the I2V models. Moreover, to obtain a stable source of dynamics, a smooth and steady camera motion in the driving video is desired. Therefore, another Vision Language Model (VLM) such as ChatGPT-4o is utilized to refine the text prompt, especially for explicit control of camera motion. After that, the refined text prompt, along with the extracted frame, is sent to the I2V models to generate a relatively high quality driving video.

However, driving video may introduce additional occlusion, disocclusion, and artifacts. To address these issues, we incorporate geometry and articulation priors from the base video as additional supervision. This helps stabilize the reconstruction, preserve geometry, and improve motion consistency in the final 4D scene. A key insight is that the original and generated sequences typically share the same first frame, enabling us to connect them through a shared canonical scene. As such, we rewind the base video (i.e., play it backward), concatenate it with the driving sequence, and perform joint optimization over the entire temporal span as shown in the second of Figure 2 . After optimization, we can truncate the reconstructed scene to get the final result.

This joint training strategy brings in strong geometry and motion constraints from the original clip and allows them to propagate into the edited segment via shared motion coefficients. We formally state this motivation in the following lemma and defer the detailed proof to Appendix A.

**Lemma 1.** *Let $d(t)$ denote the distance between two 3D points $x_A(t), x_B(t)$ defined via shared motion coefficients and time-dependent bases, as in Shape of Motion. Suppose that the motion bases are trained jointly on an edited sequence $[0, t_1]$ (without supervision) and an original sequence $[t_1, t_2]$ (with supervision), with the temporal smoothness regularization applied.*

*Then, the variance of $d(t)$ over $[0, t_1]$ is lower than if the model were trained on $[0, t_1]$ alone:*

$$\text{Var}_{t \in [0, t_1]}(d(t)) < \sigma_0$$

*where $\sigma_0$ is the variance from training without supervision.*

## 3.3 Occlusion-Aware Rigidity Regularization

During the Motion Initialization Stage, we achieve geometry preservation by effectively propagating geometry supervision throughout the sequence to achieve better initialization. Following the shape of motion [9], we use a low-rank decomposition of the motion bases $\{M_k(t)\}$ and the motion coefficients $\beta_i$ to fit the estimated flow of the scene. This scene flow is often noisy, especially in the driving video due to occlusions or motion artifacts. Inspired by physical-based regularization such as As-Rigid-As-Possible (ARAP) [51], we use a global Rigidity Loss to regularize the deformation. Specifically, we first construct a k-NN graph $\Omega = \{(x_i, x_j)\}$ among foreground Gaussians, and define the loss as:

$$\mathcal{L}_{Rigidity\_init} = \sum_{t=1}^{T} \sum_{\Omega_{i,j}} s_{ij} \|d(x_i(t), x_j(t)) - d(\mu_i, \mu_j)\|_1 \tag{2}$$

Here, we use the frame with the most visible points as a canonical frame. $\mu_i$ is the canonical position of the point $x_i$. $s_{ij}$ measures motion similarity based on the cosine similarity of motion coefficients

$\beta_i$:

$$s_{ij} = \frac{\boldsymbol{\beta}^\top \boldsymbol{\beta}_j}{\|\boldsymbol{\beta}_i\|\|\boldsymbol{\beta}_j\|} \tag{3}$$

Once the motion bases and coefficients are reasonably initialized, the object's appearance will be refined during the Photometric Fusion Stage. The photometric and tracking losses are used to further optimize the Gaussian parameters. To effectively propagate geometry supervision from the base sequence to occluded and inconsistent regions in the driving sequence, while avoid oversmoothing visible regions, we adapt the Rigidity Loss to this occlusion-aware scenario. The intuition is that the occluded regions in the driving video should correspond to visible regions in the base sequence under a rigid transformation, while the visible part should rely more on the photometric information.

We first estimate the invisibility score $\zeta_i(t) \in [0, 1]$ for each point $x_i(t)$, based on its depth difference from the rendered depth buffer $\hat{D}(t)$. The depth buffer is computed similar to color rendering as:

$$\hat{D}(t) = \sum_{i \in N} d_i(t)\alpha_i T_i, \text{ where } T_i = \prod_{j=1}^{i-1}(1 - \alpha_i) \tag{4}$$

where $d_i(t)$ is the depth of each splat $x_i(t)$ from the camera. We then define a smooth step function over the depth difference $\delta(t) = d_i(t) - \hat{D}(t)$ to compute invisibility

$$\zeta_i(t) = \begin{cases} 0 & \text{if } \delta < \tau_0 \\ 3(\frac{\delta-\tau_0}{\tau_1-\tau_0})^2 - 2(\frac{\delta-\tau_0}{\tau_1-\tau_0})^3 & \text{if } \tau_0 \leq \delta < \tau_1 \\ 1 & \text{if } \delta \geq \tau_1 \end{cases} \tag{5}$$

where $\tau_0$ and $\tau_1$ is a pre-defined boundary between visible and invisible part. This formulation allows a soft occlusion mask to guide the selective regularization of rigidity.

Finally, the occlusion-aware Rigidity loss is applied during refinement as:

$$\mathcal{L}_{Regidity\_refine} = \sum_{t=1}^{T} \sum_{\Omega_{i,j}} \zeta_i(t)\zeta_j(t)s_{ij}\|d(x_i(t), x_j(t)) - d(\mu_i, \mu_j)\|_2 \tag{6}$$

This encourages local rigidity only in regions that are not directly supervised, improving geometry propagation in occluded areas without compromising visible-region quality. We observe that overly strong rigidity constraints may encourage motion coefficient sparsity and suppress high-frequency motion. To address this, we increase the number of motion bases to 50–100 in our implementation, which empirically balances rigidity and expressiveness.

### 3.4 Disocclusion Backtracing

While the Rigidity Loss during initialization helps align the edited region motion with the original sequence, we observe that newly disoccluded parts that appear only in the driving video are hard to recover, since they were never visible in the base video and thus absent from the canonical frame.

To address this, we propose a disocclusion backtracing mechanism to recover such missing geometry by supplementing the canonical representation after initialization. We examine the visibility $\nu_i(t)$ of each tracked point $p_i(t)$, using the output of standard tracking models such as TAP-AIR [32] or Bootstap [52]. A point is considered disoccluded at time $t \in T_{\text{edit}}$ if it becomes visible in the edited video but was occluded $\mathcal{D}$ in the canonical frame:

$$v_i(t) \wedge \neg v_i(t_{cano}) \implies p_i(t) \in \mathcal{D}(t) \tag{7}$$

For each such disoccluded point, we insert a new Gaussian splat $x_i'(t)$ at its 3D location and assign its color directly from the input. We then interpolate its motion coefficients $\beta_i'$ from nearby Gaussians and backtrace the point to the canonical frame by using the inverting of Equation 1:

$$\mu_i' = \gamma(\sum \beta_{ik}' \cdot M_k(t))^{-1}x_i'(t), \forall p_i(t) \in \mathcal{D}(t) \tag{8}$$

This complements the occlusion-aware rigidity regularization by explicitly recovering disocclusion, allowing both occluded and disoccluded regions to be consistently aligned in the canonical space.

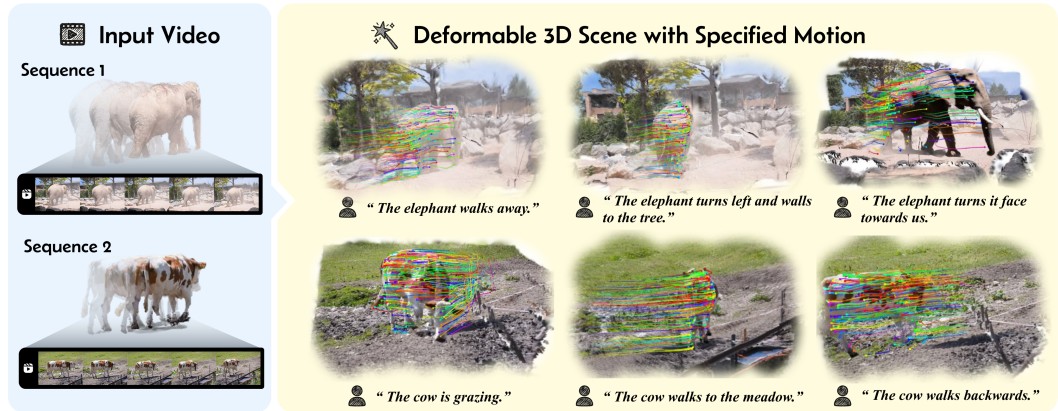

Figure 3: Samples of 4D motion creation. For the same input base video, we can create different motion variation based on the input text prompt, while preserving visual context in the base video.

## 4 Experiments

We evaluate our pipeline on both curated and in-the-wild video datasets, including DAVIS [12] and PointOdyssey [13]. For 4D restaging evaluation, we use 20 video sequences from the DAVIS dataset. Following the pipeline described in the previous section, we apply text-driven video generation to reanimate the input videos and reconstruct them using our geometry-preserving framework. To evaluate geometric consistency, we further test our method on the PointOdyssey dataset. We apply our full pipeline to reconstruct dynamic scenes and compare the resulting 3D trajectories with the ground-truth object tracking annotations, demonstrating our method's ability to preserve accurate geometry. The DAVIS sequences used in our experiments will be released to facilitate future research. All experiments are conducted on a single NVIDIA A100-SXM4 with 15GB GPU RAM. Each sequence typically takes 40 minutes for a sequence with 100 frames on 500 training epochs.

### 4.1 Reanimation and Geometry Preserving

We evaluate our pipeline in the DAVIS data set to demonstrate its ability to preserve geometry in the 4D reconstruction task. Given an input video and a text prompt describing a new motion, we use a pre-trained video generation model (e.g., Sora) to reanimate the object and then reconstruct the 4D scene using our pipeline. As shown in Figure 3, our method can generate a variation of motion based on a different text prompt. The restaged video presents novel motion while keeping the context of the input scene. Please refer to the supplementary for the input sequence and motion prompts.

A key challenge in 4D restaging is occlusion, as parts of the object may become invisible due to the new motion trajectory. Figure 4 shows the visualization for the restaged 4D scene in different views and time steps. Significant improvement in quality can be observed for occluded regions.

To assess geometry consistency, we introduce three metrics: (1) CLIP [53]-based object-centered view (OCV) consistency, (2) volume consistency, and (3) edge length stability. Existing 3D quality metrics such as PointSSIM [54] and GraphSSIM [55] are not suitable for our setting, as they are sensitive to spatial deformation and assume static correspondence. Since our pipeline explicitly models deformation, such frame-level metrics fail to capture global geometry stability. For OCV CLIP, we render each scene from a fixed relative viewpoint to the object (*e.g.*, behind or side) and compute the CLIP similarity to a clean reference frame. This measures whether the occluded geometry is preserved and visually plausible. Volume consistency $C_v$ is computed as:

$$C_v = \left( -\log \left( \frac{1}{|T_{\text{driving}}|} \sum_{t \in T_{\text{driving}}} (V_t - \bar{V})^2 \right) \right)^{\gamma}, \quad \gamma = 1.5 \tag{9}$$

where $V_t$ is the volume of the foreground voxelized at time $t$, and $\bar{V}$ is the mean volume over the driving clip. Higher values indicate lower variance and stronger volume conservation. Similarly, we measure the frame-to-frame consistency of local geometry by sampling 1000 Gaussians and tracking the standard deviation of their edge lengths to neighbors over time. Our proposed metrics better reflect the consistency of deformable 3D reconstructions under the 4D restaging task.

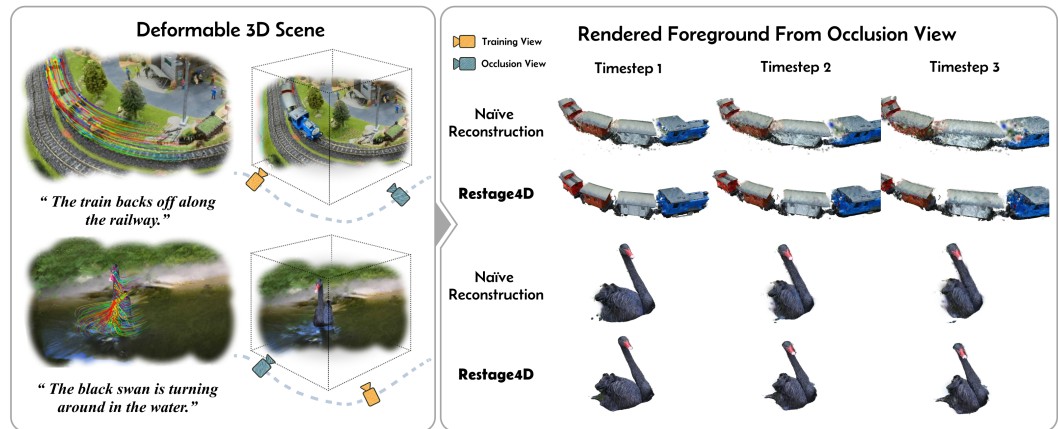

Figure 4: Samples of geometry preserving for occlusion view. For each restaged 4D scene, we visualize the foreground observing from the occlusion view at different time step. The geometry of the object can be better preserved using our method, compared to naive reconstruction.

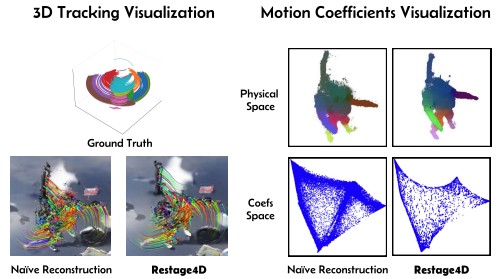

Figure 5: Visualization on PointOdessy dataset.

Table 4: Quantitative measure on PointOdessy Dataset. Restage4D can achieve higher tracking quality with geometry-preserving strategy.

| Model | 3D Tracking Loss |
|---|---|
| All | **0.05** |
| w/o Backtracing | 0.06 |
| w/o Rigidity | 0.09 |

Table 1 summarizes the results. These results reflect our core motivation for designing the occlusion-aware ARAP. While the naive ARAP improves geometric consistency metrics (Vol./Edge Consistency) for the occluded part, it applied a strong rigidity constraint globally, which significantly degrades the reconstruction quality (PSNR drops by 5.24). This is especially problematic in regions with high-frequency motion, where excessive regularization results in visually unnatural and frozen movements, leading to incorrect motions. Through empirical observations, we found a trade-off: (1) Enforcing ARAP uniformly helps preserve occluded geometry, but overly constrains visible regions and suppresses natural motion. (2) Reducing regularization improves visible motion, but sacrifices occluded region stability.

By providing a base video, we enable the model to transfer accurate articulations and geometry. Combined with selective ARAP regularization, our approach optimizes the trade-off between geometric consistency and motion fidelity. Our method achieves the highest PSNR on the training views. By leveraging priors from the base video, it also achieves strong consistency scores in terms of CLIP-score and occlusion geometry, demonstrating its ability to maintain perceptual coherence.

In Table 2, we also compare our model with other 4D reconstruction baselines on generated video condition on videos from DAVIS [12]. It's hard to get ground truth for novel view in monocular video, so we conduct additional evaluation using the Neural3DV [56], which provides 20 calibrated camera views of dynamic scenes, from which we can derive ground truth for novel views. Please refer to the appendix for construction details. We integrate other image-to-video(I2V) models into our pipeline, and it works on multiple models. In Table 3, we evaluated CausVid [57] and LTX-Video [58] for video generation and compared them with Sora. We applied our restaging pipeline to two sequences: cat-jump and cows-graze, using each I2V model as the video generation component. These results demonstrate that our Restage4D framework performs effectively across diverse I2V backbones, with consistent reconstruction quality and visual plausibility.

Table 1: Ablation study on 20 DAVIS sequences. We report the mean $\pm$ std of PSNR, SSIM and LPIPS for foregrounds, as well as CLIP similarity for occlusion views (OCV), and geometry consistency metrics. Improvements ($\Delta$) are computed per-sequence relative to the Baseline. We also evaluated the effect of using naive ARAP and occlusion-aware ARAP (o-ARAP).

| Method | PSNR ↑ | SSIM ↑ | LPIPS ↓ | OCV CLIP ↑ | Vol. Consist ↑ | Edge Consist ↑ |
|---|---|---|---|---|---|---|
| Baseline | 26.71 ± 1.97 | 0.9765 ± 0.004 | 0.0994 ± 0.049 | 81.68 ± 4.91 | 7.41 ± 3.80 | 12.66 ± 3.01 |
| w/o Base Video | | | | | | |
| + Naive ARAP | 23.67 ± 1.78 | 0.9581 ± 0.007 | 0.1733 ± 0.091 | 84.78 ± 4.39 | 9.14 ± 3.31 | 15.91 ± 3.84 |
| Δ from Baseline | -3.04 | -0.0175 | +0.0739 | +3.1 | +1.73 | +3.25 |
| + o-ARAP | 26.80 ± 2.26 | 0.9812 ± 0.006 | 0.0754 ± 0.044 | 82.65 ± 4.29 | 7.92 ± 3.97 | 13.64 ± 3.38 |
| Δ from Baseline | +0.19 | +0.0056 | -0.024 | +0.97 | +0.51 | +0.98 |
| w/ Base Video | | | | | | |
| w/o ARAP | 26.58 ± 2.21 | 0.9614 ± 0.004 | 0.1429 ± 0.063 | 82.29 ± 3.58 | 8.50 ± 3.81 | 13.24 ± 3.54 |
| Δ from Baseline | –0.13 | -0.0152 | +0.0435 | +0.61 | +1.09 | +0.58 |
| + Naive ARAP | 21.47 ± 1.94 | 0.9094 ± 0.005 | 0.1917 ± 0.083 | 87.31 ± 3.05 | 11.43 ± 4.27 | 17.30 ± 3.95 |
| Δ from Baseline | -5.24 | -0.0671 | +0.0923 | +5.63 | +4.02 | +4.64 |
| + o-ARAP | 27.12 ± 2.27 | 0.9832 ± 0.006 | 0.0632 ± 0.042 | 85.40 ± 4.11 | 10.32 ± 4.23 | 16.28 ± 3.10 |
| Δ from Baseline | +0.41 | +0.0067 | -0.0362 | +3.72 | +2.91 | +3.62 |

Table 2: Quantitative measure on PointOdessy Dataset. Restage4D can achieve higher tracking quality with geometry-preserving strategy.

| Method | DAVIS [12] | Neural3DV [56] |
|---|---|---|
| Vidu4D [59] | 19.77 | 14.74 |
| SC-GS [42] | 22.69 | 16.71 |
| Ours | **27.38** | **17.78** |

Table 3: Reconstruction PSNR using different video generation models. Restage4D can create content of similar quality using different models.

| I2V Model | cat-jump | cows-graze |
|---|---|---|
| Sora [59] | 26.74 | 26.25 |
| CausVid [42] | 23.68 | 26.45 |
| LTX-video | 25.08 | 28.05 |

## 4.2 3D Tracking

The advantages of geometry preservation are also demonstrated in 3D tracking. We evaluate our pipeline on DAVIS and PointOdyssey to show that our occlusion-aware rigidity constraint improves the tracking quality of occluded regions, As shown in the left part of Figure 5, our method can faithfully track points that become occluded during motion, as well as obtain smoother tracking trajectories, while the baseline without rigidity loss fails to maintain consistent trajectories and geometry for those regions. PointOdyssey provides ground-truth 3D point tracks, allowing us to quantitatively evaluate tracking accuracy. We compute the L1 error between the predicted and ground-truth 3D trajectories. As shown in Table 4, incorporating the occlusion-aware rigidity loss and back-tracing mechanism leads to lower tracking error, particularly in previously occluded areas, confirming its effectiveness in maintaining geometric consistency over time. We further visualize the learned motion coefficients in the right part of Figure 5. With loss of occlusion-sensitive rigidity, the coefficients become sparser and exhibit smoother spatial transitions. Occluded regions are encouraged to share similar motion coefficients with their neighbors, which helps to enforce local rigidity and preserve geometry through occlusion. This in turn improves the stability of 3D trajectories over time, leading to better tracking quality for both visible and invisible parts of the object.

## 4.3 Automatic Geometry Correction

Since our motion editing pipeline relies on videos generated by image-to-video models, it is common for these synthesized sequences to contain incorrect geometry or motion artifacts. This highlights an additional advantage of incorporating the original video as an auxiliary input during training: by using accurate observations as supervision, the model can automatically correct geometric and motion-related errors present in the generated video. As shown in Figure 6, our method successfully fixes several failure cases introduced by the video generation model. In the first example, the model mistakenly attaches a road sign to the bus, corrupting the appearance of the foreground. In the second case, a part of the background is erroneously fused to the camel's hump, and the motion of its legs is incorrectly changed. Through joint training with the original sequence, Restage4D is able to correct these inconsistencies, recovering plausible appearance and producing more realistic deformation,

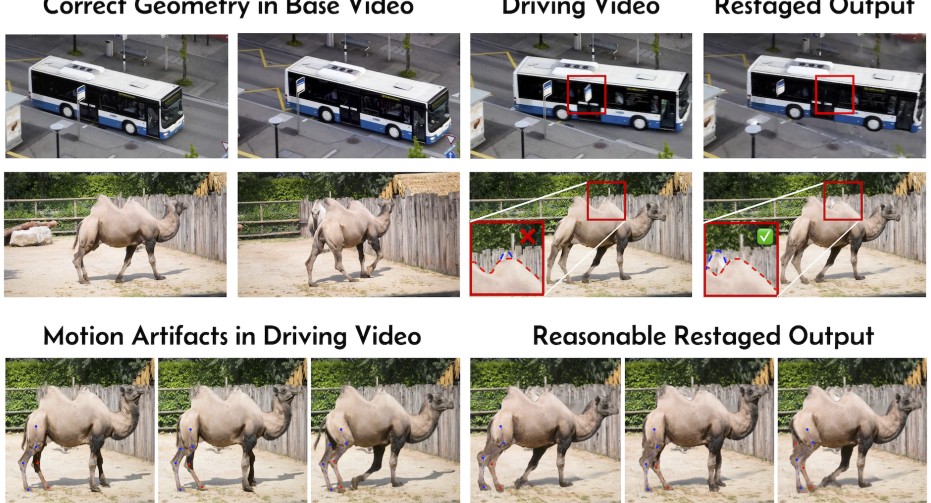

Figure 6: Samples of geometry and motion correction. Conditioning on the base video, Restage4D can use the learned articulation to correct some appearance and motion artifacts in the driving video.

even from flawed motion inputs. These results demonstrate that leveraging accurate prior observations from the original sequence not only improves geometric consistency, but also enables automatic correction of hallucinated artifacts and implausible motion, bridging the gap between generative and physically faithful reconstructions.

## 5 Conclusion and Limitation

In this work, we introduce **4D restaging**, a new video-conditioned 4D content creation task that aims to reanimate deformable 3D reconstructions from a single real-world video. By combining a pre-trained video generation model with our geometry-preserving video-rewinding training pipeline, Restage4D enables the transfer of motion dynamics from a synthetic video while preserving the physical plausibility and geometric fidelity of the original scene. Our approach leverages the input video as a source of shape and articulation priors, allowing us to generate high-quality 4D scenes that remain consistent even in occluded or disoccluded regions.

However, our method has limitations. If the video generation model produces sequences with severe geometric artifacts or unrealistic motions, the resulting supervision may not be sufficient to recover accurate 4D geometry. Additionally, while we leverage geometry supervision from real videos, we assume the input object remains temporally consistent across sequences, which may not hold for highly deformable or textureless objects. Our framework involves generative video models, which may pose risks when applied to human-centric content, potentially leading to the synthesis of misleading or harmful videos. One possible solution is incorporating prompt filtering mechanisms during the prompt refinement stage to detect and block inappropriate or sensitive prompts.

We hope that our work offers insights into how real-world video properties (*e.g.*, geometry priors, articulation, visibility) can be used to stabilize and guide 4D content creation, bridging generative modeling with reconstruction-driven pipelines.

## 6 Acknowledgement

This work is supported by the Intelligence Advanced Research Projects Activity (IARPA) via Department of Interior/ Interior Business Center (DOI/IBC) contract number 140D0423C0074. The U.S. Government is authorized to reproduce and distribute reprints for Governmental purposes notwithstanding any copyright annotation thereon. Disclaimer: The views and conclusions contained herein are those of the authors and should not be interpreted as necessarily representing the official policies or endorsements, either expressed or implied, of IARPA, DOI/IBC, or the U.S. Government. This work is also supported in part by a Sony Faculty Award.

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

# A    Proof of Lemma 2

We introduce Lemma 2 to demonstrate the necessity of using base video as additional input, which is

**Lemma 2.** *Let $x_A(t), x_B(t) \in \mathbb{R}^3$ be the positions of two 3D points at time $t \in [0, T]$, defined by a shared time-independent motion coefficient vector $\beta \in \mathbb{R}^K$, and a set of time-dependent, spatially smooth motion bases $\{M_k(t)\}_{k=1}^{K}$.*

*Assume the following:*

*(i) The scene is trained over two segments: an edited clip $[0, t_1]$ without supervision, and an original clip $[t_1, t_2]$ with ground-truth geometry supervision;*

*(ii) A temporal smoothness regularization is applied on the motion bases $M_k(t)$, i.e., $\sum_{k,t} \|M_k(t+1) - M_k(t)\|^2$;*

*(iii) Each point's position at time $t$ is given by a transformation composed from the bases and coefficients:*

$$x(t) = \sum_{k=1}^{K} \beta_k \cdot M_k(t)(x_0),$$

*where $x_0$ is the canonical position.*

*Then, the variance of the pairwise distance $d(t) = \|x_A(t) - x_B(t)\|$ within the edited clip satisfies:*

$$\sigma_2 := Var_{t \in [0, t_1]}(d(t)) < \sigma_0,$$

*where $\sigma_0$ is the variance obtained by training only on the edited clip without supervision.*

We can prove this lemma using

*Proof.* Let $x_A^0, x_B^0 \in \mathbb{R}^3$ be fixed canonical positions of two points. At time $t$, their positions in world coordinates are given by:

$$x_i(t) = \sum_{k=1}^{K} \beta_k \cdot M_k(t)(x_i^0), \quad \text{for } i \in \{A, B\},$$

where $\beta \in \mathbb{R}^K$ is shared and fixed, and $M_k(t) : \mathbb{R}^3 \to \mathbb{R}^3$ are time-varying motion basis functions.

Define the pairwise squared distance:

$$d^2(t) = \|x_A(t) - x_B(t)\|^2 = \left\| \sum_{k=1}^{K} \beta_k \cdot \big( M_k(t)(x_A^0) - M_k(t)(x_B^0) \big) \right\|^2.$$

Let $\Delta_k(t) := M_k(t)(x_A^0) - M_k(t)(x_B^0) \in \mathbb{R}^3$, then:

$$d^2(t) = \left\| \sum_{k=1}^{K} \beta_k \cdot \Delta_k(t) \right\|^2.$$

This is a quadratic form in the temporal functions $\Delta_k(t)$, linearly combined by fixed weights $\beta_k$. Its time-variance depends on the temporal variability of $\Delta_k(t)$.

Now, consider the regularized training setup:

  (i) The motion bases $\{M_k(t)\}$ are supervised only in $[t_1, t_2]$, enforcing accurate deformation there;

  (ii) A temporal smoothness regularization is imposed:

$$\mathcal{L}_{\text{smooth}} = \sum_{k=1}^{K} \sum_{t} \|M_k(t+1) - M_k(t)\|^2.$$

Due to this regularization, each $M_k(t)$ evolves smoothly over time. Let us denote the discrete temporal second difference of $\Delta_k(t)$ as:

$$\delta_k(t) := \Delta_k(t+1) - \Delta_k(t).$$

Then, smoothness implies that $\|\delta_k(t)\|$ is small, especially near the boundary $t = t_1$, where $M_k(t)$ is influenced by supervision at $t > t_1$. Since the functions $\Delta_k(t)$ are more constrained in this case (compared to unregularized training), their fluctuations in $[0, t_1]$ are suppressed.

Let us define:

$$f(t) := \sum_{k=1}^{K} \beta_k \cdot \Delta_k(t), \quad \text{then} \quad d^2(t) = \|f(t)\|^2.$$

The temporal variance of $d(t)$ satisfies:

$$\text{Var}_{t \in [0, t_1]}(d(t)) = \text{Var}_{t \in [0, t_1]}(\|f(t)\|).$$

Since $f(t)$ is a fixed linear combination of smoother functions $\{\Delta_k(t)\}$, its temporal fluctuation is reduced by the regularization. That is:

$$\text{Var}_{t \in [0, t_1]}(\|f(t)\|) \downarrow \quad \text{as} \quad \mathcal{L}_{\text{smooth}} \downarrow.$$

In contrast, training on $[0, t_1]$ alone without regularization leads to unconstrained $M_k(t)$, and thus larger temporal variability in $f(t)$. Therefore:

$$\sigma_2 := \text{Var}_{t \in [0, t_1]}(d(t)) < \sigma_0,$$

where $\sigma_0$ is the variance from training without supervision or smoothness.

$\square$

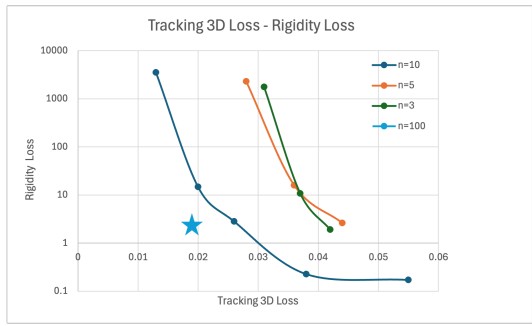

Figure 7: Effect of changing the number of bases.

# B  Training Details

## B.1  2D Priors

To enable 4D reconstruction from a monocular video, we leverage a set of 2D foundation models to provide reliable initialization. Following the pipeline of Shape-of-Motion, we use Segment Anything for foreground segmentation and Track Anything to propagate the masks across frames. For camera motion and point correspondence, we adopt Tapir for 2D point tracking and MegaSAM for estimating camera poses. Additionally, we incorporate dense depth maps from VideoDepthAnything to provide richer geometric details for the scene.

## B.2  Hyperparameters

To achieve geometry-preserving reconstruction while maintaining high-frequency motion details, we found it essential to carefully tune the rigidity loss. Since the overall motion is primarily determined during the motion initialization stage, we evaluate the effect of different combinations of rigidity loss weights and the number of motion bases.

As shown in Figure 7, each curve represents the trade-off between object rigidity and tracking accuracy for a given number of motion bases, as we vary the rigidity loss weight from 0 to $10^{-1}$. Our results show that increasing the number of motion bases can improve this trade-off curve, and that a loss weight of $10^{-3}$ offers a good balance. Based on this analysis, we use 100 motion bases and set the rigidity loss weight to $10^{-3}$ in all experiments.

# C  Samples from DAVIS Dataset

To qualitatively evaluate our 4D restaging pipeline, we present several representative samples from the DAVIS dataset in Figure 8. For each sequence, we show: (1) two keyframes from the base video (original monocular input); (2) the corresponding motion prompt describing the desired new motion; and (3) two keyframes from the generated driving video, synthesized using a pre-trained image-to-video model (*e.g.*, Sora) conditioned on the prompt and the first frame. These examples demonstrate the diversity of motion prompts supported by our pipeline, including changes in direction, speed, and posture. They also highlight the realism and coherence of the generated driving video, which serves as the motion source for our 4D reconstruction. For each example, we visualize the base and drive videos using two keyframes (top and bottom) to illustrate the overall temporal dynamics.

# D  Evaluation on Neural 3D Video Synthesis dataset

Evaluating the novel view by comparing with ground truth is important for validating view generalization in monocular reconstruction, but the groundtruth is hard to get. To address this, we conduct additional evaluation using the Neural 3D Video Synthesis dataset, which provides 20 calibrated camera views of dynamic scenes, from which we can derive ground truth for novel views We adopt the following protocols:

To simulate a monocular causal video with natural camera motion, we perform depth-first search (DFS) to find a smooth camera trajectory across the 20 views. We then sample 100 frames along

this path to construct a pseudo-monocular video. We treat the first 60 frames as the base video and the remaining 40 frames as the driving video. For evaluation, we select a camera that is not on the sampled path and render from this novel viewpoint, which has ground truth available for comparison. We compute PSNR scores for three training strategies: (i) training only on the generated video, (ii) joint training with the base video, and (iii) joint training with ARAP refinement.

## E  More Results

To demonstrate generalization of the pipeline, we also provide visualization on some web-collected sequences shown in Figure 3.

## F  Video Demo

Please refer to the video demo in the attachment.

| Base Video | Prompt | Drive Video |
|:----------:|:------:|:-----------:|

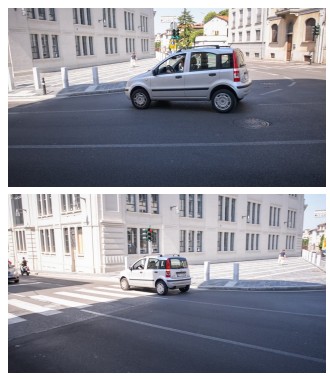

"A small silver car slowly drives up onto the sidewalk from the street in a quiet urban area. The camera stays fixed in position but smoothly pans to follow the vehicle's motion as it mounts the curb and continues onto the pedestrian pavement. Surrounding buildings are modern and light-colored, with crosswalk lines, bollards, and a few pedestrians in the background. The motion feels slightly unusual yet calm, with no abrupt changes in camera movement."

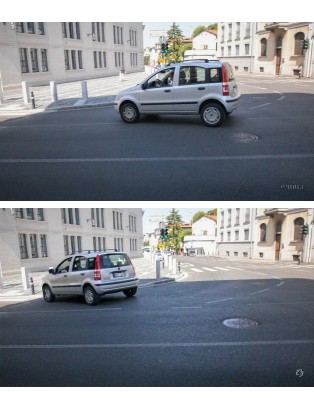

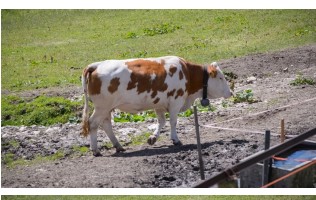
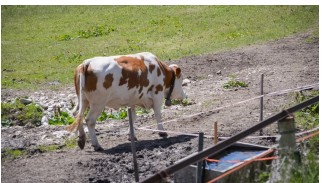

"A brown and white cow slowly turns its body and walks toward the grassy field to graze. The camera remains fixed in physical position but gently pans to follow the cow's movement as it turns and begins eating grass. The scene takes place in a quiet rural pasture under bright daylight, with patches of soil, stones, and fencing visible in the foreground. The motion is natural and unhurried, with the cow staying centered in the frame."

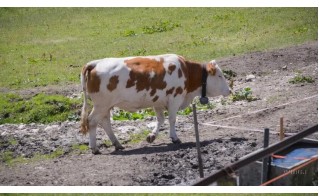
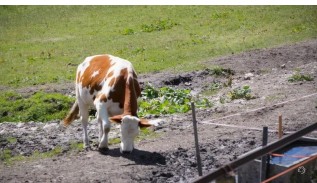

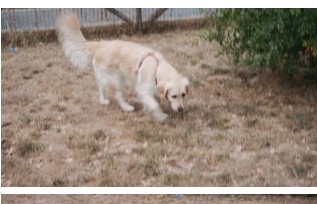
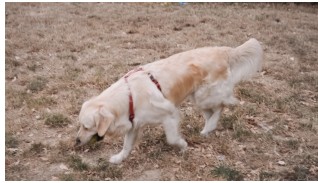

"A light-colored dog wearing a harness suddenly dashes into the nearby bushes on the right side of the frame. The camera remains completely still, capturing the dog's quick movement as it disappears into the foliage. The setting is a dry grassy area with sparse leaves, and a fence and road are visible in the background. The scene feels spontaneous and natural, with no camera tracking or zooming."

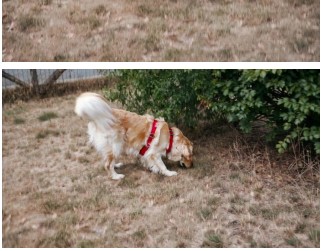

Figure 8: Examples of motion prompts and corresponding base/drive videos. Each column shows the input video, the motion prompt, and the generated video. Each video is visualized using two keyframes (top and bottom).

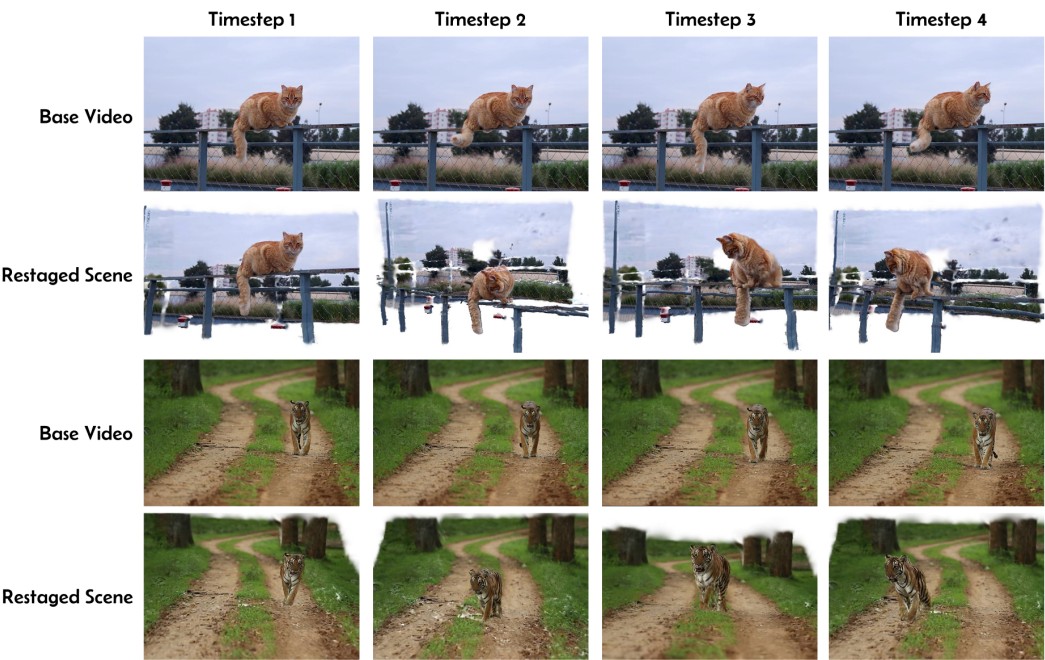

Figure 9: Some samples of 4D restaging on in-the-wild video, which are collected from Internet.

