# OpenReview forum: "Restage4D: Reanimating Deformable 3D Reconstruction from a Single Video"
_NeurIPS.cc/2025/Conference — NeurIPS 2025 poster_

### Official Review · Reviewer_Nx6n · 2025-06-30

**Clarity:** 3
**Significance:** 2
**Originality:** 3
**Rating:** 4
**Confidence:** 4

**Summary:**

This work introduces a new task, video restaging, and builds a novel framework to achieve controllability and geometric fidellity in 4D reconstruction. The task is to generate new motion for the object in input video. The main idea of this work is to re-animate the first frame of input video and leverage base video to correct the artifacts from generative videos during 4d reconstruction. Video-rewinding joint training strategy is proposed with occlusion-aware ARAP and disocclusion

**Questions:**

Were the generative videos cherry-pick? If so, what was the picking criteria?

**Ethical Concerns:**

["NO or VERY MINOR ethics concerns only"]

**Final Justification:**

The authors have provided extensive experiment results, both comparison and ablation, to address my concerns. So I decide to raise the recommendation.

**Limitations:**

Yes

**Quality:**

2

**Strengths And Weaknesses:**

strength:
1. this work propose a novel task, re-animating 4D object in the wild
2. The paper is easy to follow, and the result is good.

weakness:
1. lack comparison the state-of-the-art 4D reconstruction methods, e.g., shape-of-motion (the baseline in this work), SCGS and Vidu4D. It is suggested to take sololy new video as input for these methods for comparison.
2. The evaluation presented in Table 1 is conducted solely on training views, which is not a reasonable approach. It would be more appropriate to evaluate novel view synthesis capabilities. While I acknowledge that generated videos lack ground truth for novel views, the authors could leverage public multi-view video datasets for evaluation, such as the Neural 3D Video Synthesis dataset[1] and Dycheck[2]. A possible evaluation strategy would be to split these videos into two parts: using the first segment as the base video and the second as the reference for generated content. Although this approach would yield artifact-free "generated" videos, it's worth noting that state-of-the-art video generation models can also produce artifact-free results after careful cherry-picking.
[1] Neural 3D Video Synthesis from Multi-View Video
[2] Monocular Dynamic View Synthesis: A Reality Check

If my concerns are adequately solved during rebuttal, I'm glad to raise my assessment.

---

> ### Author Rebuttal · Authors · 2025-07-31
>
> We appreciate reviewer `Nx6n` finds our proposed task novel and the result good. We address the concerns with additional experiments.
>
> ## [Weakness 1] Comparison with Other Models
>
> We have conducted comparisons between Restage4D and two recent methods, SC-GS and Vidu4D, on generated sequences from the DAVIS dataset. Specifically, we use two videos: horsejump-high and flamingo. For each, we apply a motion prompt and evaluate reconstruction quality both qualitatively and quantitatively via PSNR.
> - *Sequence 1*:
>   - *Input Video: A rider jumps a horse over an obstacle.*
>   - *Text Prompt: "The horse turns around."*
>   - *Reconstruction Result:*
>       - *SC-GS fails to model realistic deformation; the horse remains static, with rotating artifacts.*
>       - *Vidu4D reconstructs the foreground reasonably well but lacks background and suffers from blurred edges.*
>       - ***Ours** successfully reconstructs the full scene (foreground and background) with realistic deformation and consistent motion.*
>
> - *Sequence 2*:
>   - *Input Video: A flamingo drinks water with head motion.*
>   - *Text Prompt: "The flamingo moves its leg up and down."*
>   - *Reconstruction Result:*
>       - *SC-GS fails to animate the leg, which remains static. The entire flamingo even disappears due to camera motion in some frames.*
>       - *Vidu4D only reconstructs the foreground with a gray background. The flamingo follows the prompt.*
>       - ***Ours** produces a full-scene reconstruction with consistent leg motion and high visual quality.*
>
> **Quantitative Results (PSNR)**:
> | Method | horsejump-high-turn | flamingo-leg |
> | -------- | -------- | -------- |
> | SC-GS     | 22.41     | 21.57     |
> | Vidu4D     | 24.2 (foreground)     | 26.4 (foreground)      |
> | **Ours**     | **25.8**     | **28.91**     |
>
> These results show that:
>
> 1.	SC-GS, while effective on synthetic multiview datasets (e.g., D-NeRF [citation]), struggles with monocular, real-world video due to occlusion and motion complexity.
> 2.	Vidu4D focuses solely on reconstructing foregrounds and lacks full-scene fidelity.
> 3.	Restage4D handles deformable motion and scene-level reconstruction holistically, achieving higher PSNR and better visual quality under diverse motion prompts.
>
> We will include the comparison in our final version.
> ## [Weakness 2] Evaluation
> We agree that evaluating the novel view by comparing with ground truth is important for validating view generalization in monocular reconstruction.
> To address this, we conduct additional evaluation using the Neural 3D Video Synthesis dataset [1], which provides 20 calibrated camera views of dynamic scenes, from which we can derive ground truth for novel views
> We adopt the following protocols:
>
> 1.	To simulate a monocular causal video with natural camera motion, we perform depth-first search (DFS) to find a smooth camera trajectory across the 20 views. We then sample 100 frames along this path to construct a pseudo-monocular video.
> 2.	We treat the first 60 frames as the base video and the remaining 40 frames as the driving video.
> 3.	For evaluation, we select a camera that is not on the sampled path and render from this novel viewpoint, which has ground truth available for comparison.
> 4.	We compute PSNR scores for three training strategies: (i) training only on the generated video, (ii) joint training with the base video, and (iii) joint training with ARAP refinement.
>
>
> We evaluate on two sequences: sear-steak and cut-roasted-beef. The PSNR results are as follows:
>
> |  | sear-steak | cut-roasted-beef |
> | -------- | -------- | -------- |
> | Driving Video   |   12.87 ± 0.37   |  17.84 ± 0.56    |
> | Joint     |   17.18 ± 0.56   |  19.14 ± 0.63  |
> | Joint + ARAP     | **17.70 ± 0.46**  |   **19.56 ± 0.72**  |
>
> These results demonstrate the benefits of using the base video as supervision and incorporating geometry-aware refinement. Joint training notably improves generalization to unseen viewpoints, and the addition of ARAP yields further gains.
>
> ## [Question] Criteria to select video
>
> Currently, the generated driving videos are manually filtered to ensure basic quality:
>
> 1.	Camera: We discard sequences with hard cuts or abrupt viewpoint changes, as these cause failures in dense point tracking and violate the motion consistency assumption.
> 2.	Motion: We also exclude sequences where the motion does not align with the input text prompt, to ensure edit fidelity.
>
> We are exploring the use of vision-language models (VLMs) to automate this filtering process, and plan to incorporate it into our pipeline in future work.

---

> > ### Comment · Reviewer_Nx6n · 2025-08-03
> > **Question**
> >
> > Thanks for the authors' effort to address my concerns. I have some follow-up questions:
> > 1. Comparison: I suggest to evaluate on more sequences to further demonstrate the state-of-the-art performance of the proposed method. Two sequences is not enough.
> > 2. Please provide the result of comparison baselines on Neural3DV dataset, using all sequences if possible

---

> > > ### Author Response · Authors · 2025-08-04
> > >
> > > Thank you for the follow-up. To better compare against baseline methods, we evaluated Vidu4D and SC-GS on 9 sequences from the reanimated DAVIS dataset. Due to the time constraints required to preprocess inputs for both baselines, we selected a representative subset. The PSNR results are summarized below
> > >
> > >
> > >
> > > | Reanimated Sequence | Vidu4D | SC-GS | Ours |
> > > | -------- | -------- | -------- | -------- |
> > > | `horsejump-high-turn`     | $24.2$   | $22.41$     | **25.6**    |
> > > | `rhino-turn`     | $16.31$     | Failed$^1$     | **29.4**   |
> > > | `flamingo-walk`     | $26.4$     | $21.75$     | **28.91**     |
> > > | `train-back`     | $16.05$     | $22.67$     | **26.4**     |
> > > | `dog-grass`     | $19.14$     | $23.7$     | **26.96**     |
> > > | `elephant-face`     | $22.32$     | Failed$^{2}$     | **30.48**     |
> > > | `blackswan-swim`     | $17.29$     | $23.12$     | **27.83**     |
> > > | `camel-look`     | $20.01$     | Failed$^{2}$     | **24.4**     |
> > > | `cows-around`     | $16.28$     | Failed$^1$     | **26.52**     |
> > > | Average     | $19.77\pm 3.38$     | $22.69\pm 0.65$     | **27.38 $\pm$ 1.73**     |
> > >
> > > $^1$ Failed due to COLMAP's inability to register frames from limited camera motion.
> > > $^2$ Failed during control point initialization, resulting in zero valid points.
> > >
> > > As shown above, both Vidu4D and SC-GS struggle under the monocular, causal video setup, where even the training-view PSNR falls significantly below our method.
> > >
> > > We also evaluated all three methods on all six sequences from the Neural3DV dataset, using the same settings as described in our previous response (holding out one view for as ground truth):
> > >
> > > | Reanimated Sequence | Vidu4D | SC-GS | Ours |
> > > | -------- | -------- | -------- | -------- |
> > > | `sear-steak`     | $13.15$   | $15.48$     | **17.7**     |
> > > | `cut-roasted-beef`     | $13.09$     |$16.15$     | **19.56**   |
> > > | `coffee-martini`     | $16.78$     | $18.21$     | $18.24$     |
> > > | `cook-spinach`     | $15.66$     | $16.76$     | **17.01**     |
> > > | `flame-steak`     | $15.92$     | $16.32$     | **17.87**     |
> > > | `flame-salmon`     | $13.85$     | **17.37**     | $16.34$     |
> > > | Average     | $14.74\pm 1.33$     | $16.71\pm 0.81$     | **17.78 $\pm$ 0.93**     |
> > >
> > > Note that even though we used a camera path to mimic monocular video, the input video still offers abundant 3D cues due to the large camera motion. Besides, Neural3DV contains limited human motion. Since our method is designed for monocular causal videos with complex motion and occlusion, the domain gap somewhat reduces the advantage of our pipeline. Nonetheless, our approach still outperforms both baselines across most sequences.
> > >
> > > We hope this addresses your concerns.

---

> > > > ### Comment · Reviewer_Nx6n · 2025-08-05
> > > >
> > > > Thanks for providing experiments to address the concerns. I'll raise my score.

---

> ### Author Response · Authors · 2025-08-02
>
> Dear reviewer `Nx6n`,
>
> Please let us know if we solved your concerns. We are happy to discuss if you have any questions. Your feedback means a lot to us.

---

### Official Review · Reviewer_duUh · 2025-06-30

**Clarity:** 4
**Significance:** 3
**Originality:** 3
**Rating:** 5
**Confidence:** 4

**Summary:**

This paper presents Restage4D, a pipeline for reanimating deformable 3D scenes by conditioning on a single real-world video. It first generates a synthetic driving video based on the input’s first frame, then the method uses the original sequence as a supervisory signal to correct artifacts from synthetic motion. The input real-world video provides the shape and articulation model, while the generated video provides new dynamics.

**Questions:**

N.A.

**Ethical Concerns:**

["NO or VERY MINOR ethics concerns only"]

**Final Justification:**

The authors' rebuttal has addressed my concerns, I will keep my positive score.

**Limitations:**

The authors have discussed the method’s limitations, but they omit the societal impact—please include it in the final version.

**Paper Formatting Concerns:**

N.A.

**Quality:**

3

**Strengths And Weaknesses:**

Strengths

1. The paper is well written and the task is highly engaging.

2. The video-rewinding joint-training scheme is both well motivated and insightful.

Weaknesses

1. All examples involve a single foreground object. How does the method handle scenes with multiple interacting objects—for instance, “two people dancing together”?

2. Some minor typos to fix: Line 176: “bas” → “base”, Video at 4:12 (bus case): the wrong prompt is shown.

3. Additional qualitative results—especially on more complex scenes—would strengthen the empirical validation of your approach.

---

> ### Author Rebuttal · Authors · 2025-07-31
>
> We thank the reviewer `duUh` for finding our task engaging and pipeline insightful. We address the concerns by providing more experiments.
>
> ## [Weakness 1] Multiple Objects
> **Our monocular reconstruction pipeline does not impose any restriction on the number of objects in the scene.**
> In practice, the use of K-Nearest Neighbor (KNN) graph construction for the ARAP loss leads to localized connectivity among nearby splats. As a result, splats naturally group into object-level clusters, since intra-object distances are smaller than inter-object distances. This behavior helps maintain per-object rigidity and avoids enforcing unrealistic deformation across different objects.
>
> At object interaction boundaries (e.g., contact surfaces between two objects), the proximity-based KNN may occasionally link splats across object boundaries. While this typically has a limited impact, it may affect fine-grained interaction modeling (e.g., relative motion at contact points). Addressing this challenge remains an important direction for future work.
> ## [Weakness 2] Typos
> We will fix all the typos in the revised version.
>
> ## [Weakness 3] More Complex Scene
> We run our restaging pipeline on a more complex scene with two interacting objects, a cat and a dog.
> - *Input video content: A dog and a cat are in the bathroom. The cat is standing on the toilet lid, and the dog is by its side on the floor. They first sniff each other, then the cat turns around at the toilet lid.*
> - *Text prompt: "The cat jumps towards the dog."*
> - *Restaged video: The video follows the prompt, and there are two sets of dynamic Gaussian Splats representing the cat and dog, respectively.*
> |  | PSNR | OCV CLIP | Vol. Consist | Edge Consist |
> | -------- | -------- | -------- | -------- | -------- |
> | Cat-jump-dog     | 31.53     | 84.27     | 8.84 |  17.25 |
> | DAVIS Sequence | 27.12 | 85.40 | 10.32 | 16.28 |
>
> This case demonstrates that our pipeline can generalize to more complex scenes with multiple objects. We will include this sample in our final version.
>
> ## [Limitation]
> **Social Impact**
> Our framework involves generative video models, which may pose risks when applied to human-centric content, potentially leading to the synthesis of misleading or harmful videos. One possible solution is incorporating prompt filtering mechanisms during the prompt refinement stage to detect and block inappropriate or sensitive prompts.

---

> > ### Comment · Reviewer_duUh · 2025-08-03
> > **Reply to authors' rebuttal**
> >
> > Dear authors, thank you for the rebuttal. My concerns have been addressed, I will keep my positive score.

---

> > > ### Author Response · Authors · 2025-08-03
> > >
> > > Thank you again for your timely reply and discussion.

---

> ### Author Response · Authors · 2025-08-02
>
> Dear reviewer `duUh `,
>
> Please let us know if we solved your concerns. We are happy to discuss if you have any questions. Your feedback means a lot to us.

---

### Official Review · Reviewer_gQQs · 2025-07-02

**Clarity:** 3
**Significance:** 2
**Originality:** 3
**Rating:** 4
**Confidence:** 3

**Summary:**

Restage4D proposes a task to generate a new 3D sequence based on a given video. It proposes a framework that first does image-to-video generation, and then reconstructs the new video together with the original video into 3D.  To do so, a video concatention strategy and occlusion-aware rigidity handling are proposed.

**Questions:**

Please see the weakness section.

**Ethical Concerns:**

["NO or VERY MINOR ethics concerns only"]

**Final Justification:**

I see value in the proposed techniques, particularly the strategies for handling occlusion and enforcing rigidity constraints. I’m convinced that these components are effective for reanimating a video in 3D space, and for that reason, I am inclined to slightly raise my score to a borderline accept.

However, I still believe the problem formulation and evaluation methodology require improvement. The current setup would make more sense if the task were framed as, for example, reconstructing multiple videos that occur concurrently.

**Limitations:**

Yes.

**Quality:**

3

**Strengths And Weaknesses:**

Strength:
1. The 4D reanimation task sounds interesting and new. It could be useful for some downstream simulations like autonomous driving and robotics tasks.
2. The paper identifies the geometry inconsistency as the challenge and tackles it with video rewinding, occlusion-aware rigidity, and occlusion backtracking, which makes a lot of sense to me. The results also show that the method is able to fix some inconsistencies in the newly generated video.

Weakness:
1. Using image-to-video generation as the first step seems a debatable design choice. The first frame of a video does not necessarily capture all important information, for example, the subject might not be present in the first frame, or textures are not observable in the first frame. To tackle this task, it seems that how to generate videos that are consistent with the input deserves more discussion. For example, we might want to let the video model take the full input video as a condition. Instead, the paper relays the inconsistency problem to the reconstruction step and tries to fix it with rigidity constraints, which I think is less optimal.

2. The task seems to be defined to be too complex and specific, with a mix of reconstruction and generation. Currently, the input is a video, and the output is an edited 4D reconstruction of the input video. It seems to involve several more fundamental tasks, like single-view video reconstruction, video editing, and 3D editing. The complex task setting also makes the evaluation difficult. In the paper, the evaluation mainly reports 3D consistency to the original video and PSNR to the generated video. First, as an editing task ("restage"), the editing accuracy is not evaluated at all. One could achieve a perfect score if the video stays identical before and after restaging. Second, evaluating PSNR to the generated video seems a bit weird, since the task does not necessarily require having a generated video as an intermediate output. Overall, my concern is if the task is significant enough to be separated topic and if the current evaluation is really suitable for it.

3. The technical part mainly makes improvements on "shape-of-motion" to better serve the purpose of reconstructing one video using the geometry of another video. All the technical parts bring solid improvements, but they are also not significantly novel or transferable to other domains.

Minor:

- L276: "We evaluate our pipeline on and PointOdyssey"

---

> ### Author Rebuttal · Authors · 2025-07-31
>
> We thank reviewer `gQQs` for finding that our new task 4d restaging is interesting and new, and the problem identification makes sense.
>
> ## [Weakness 1] Design Choice
> - **Flexible Frame Selection**
> Our pipeline does not assume that the first frame is always used for video generation. In practice, we select a frame that contains sufficient semantic and appearance information, and it can be any frame from the input sequence. For example, in our cat sequence experiment, we use the 48th frame as the reference frame to generate a video of a jumping cat in the `[Weakness 3]` part for reviewer `dprx`. We will include this example in the final version.
> - **Research Scope and Motivation**
> While we agree that leveraging the full input video during generation is a promising direction, our work focuses on the reconstruction stage, which is orthogonal and complementary to advances in video generation. Even with powerful video generation models, occlusion, disocclusion, and temporal inconsistencies are inevitable when introducing novel motion. Our insight is that these challenges can be effectively addressed during reconstruction by leveraging input video geometry and structure. This separation of concerns allows our framework to remain compatible with different generation models and improve results in a plug-and-play fashion.
>
> ## [Weakness 2] Task and Evaluation
> - **Task Necessity**
> Our goal is to explore a novel paradigm for 4D content creation, where motion is conditioned on both a reference video and a text prompt (L26–28). This goes beyond typical text-to-4D or image-to-4D settings by allowing users to control motion and appearance jointly through intuitive and accessible inputs. The video-based control provides richer motion priors and more precise specifications, making it particularly valuable for downstream applications such as scene animation, 4D data synthesis, and artistic content creation. We believe this justifies treating “restaging” as a distinct and meaningful research problem.
> - **Evaluation Metrics**
>   1. Editing Accuracy: Evaluating prompt-following behavior in generative models is still an open problem in both image and video generation. In our current setup, we mitigate this by manually filtering driving videos to ensure alignment with the text prompt. As shown in Figures 1, 3, and 4, the resulting motion diverges clearly from the base video, demonstrating successful editing rather than identity preservation.
>   2. PSNR: This metric serves a clear purpose in our reconstruction-based framework. Since the driving video is pre-filtered to match the desired motion, PSNR reflects how well the reconstructed 4D scene conforms to the intended motion and appearance. It measures the faithfulness of the reconstruction to a desired visual target, which is particularly relevant when using generative video as input. We acknowledge that more semantic-aware metrics would strengthen evaluation, and we plan to explore automatic prompt-following metrics (e.g., using vision-language models) as future work.
>
> ## [Weakness 3] Novelty and Transferability
> - **Novelty**
>   - As stated in L81–86, **our core contribution lies in introducing a new paradigm for video-conditioned 4D content creation**, where motion and geometry are jointly guided by the base video and edited via a generated driving video. While ARAP-based regularization is not new in isolation, our occlusion-aware variant is specifically tailored to tackle challenges unique to this setting, especially how to reconstruct geometry across occlusion/disocclusion boundaries introduced by new motions. Therefore, the technical novelty lies in how these components interact synergistically in our pipeline rather than in isolation.
>   - As detailed in our response to reviewer `dprx`, we visualize the effect of each module using the bear sequence from DAVIS. This example involves both occluded (bear’s face turning away) and disoccluded (right side of the bear turning visible) regions. A detailed qualitative analysis shows how each component incrementally improves performance:
>     - Synthetic-only optimization fails to recover geometry in occluded areas.
>     - Adding ARAP preserves shape better, but remains limited.
>     - Joint optimization brings the base video into the loop, enabling geometry transfer.
>     - Occlusion-aware ARAP loosens rigidity for disoccluded regions.
>     - Backtracing explicitly supplements sparse areas, speeding up convergence.
>
>     **These insights collectively demonstrate that each technical module plays a novel and indispensable role when integrated for the specific goal of controllable 4D restaging.**
>
>
> - **Transferability**
> While our pipeline is designed for restaging, the core ideas are generalizable:
>   1. Using video-conditioned articulation as a source of geometry/motion supervision.
>   2. Using occlusion-aware regularization to handle incomplete or one-sided observations.
>   3. Leveraging joint optimization with deformable priors to fuse geometry and motion across input modalities.
>   These strategies can be directly applied to tasks such as video-to-3D reconstruction, editable 4D avatar generation, and controllable NeRF editing, particularly in settings where occlusions and motion-driven deformation are challenging.
>
> ## [Minor] Typo
> We will fix all the typos in the revised manuscript.

---

> ### Author Response · Authors · 2025-08-02
>
> Dear reviewer `gQQs`,
>
> Please let us know if we solved your concerns. We are happy to discuss if you have any questions. Your feedback means a lot to us.

---

### Official Review · Reviewer_dprx · 2025-07-02

**Clarity:** 2
**Significance:** 3
**Originality:** 3
**Rating:** 5
**Confidence:** 4

**Summary:**

This paper addresses the task of video-conditioned 4D content generation. Specifically, given a video depicting an object and a target motion text, the goal is to generate novel videos that are conditioned on (1) the object's appearance from the input video and (2) motion synthesized via image-to-video generation models.

The key idea is to jointly optimize the reversed input video and the synthesized video, allowing motion coefficients to be shared across both domains.

In addition, the authors introduce an occlusion-aware rigidity regularization, which leverages the original video to recover occluded regions, as well as a disocclusion backtracing mechanism to handle newly revealed areas in the synthesized video.

**Questions:**

1) **[Clarification on Disocclusion Backtracing]**
As I understand it, frames of the synthetic video are jointly optimized during initialization, meaning that disoccluded regions should also be optimized alongside the base video. Given this, it is unclear why the disocclusion backtracing module is necessary I would appreciate it if the authors could elaborate on why these regions are not sufficiently addressed through joint optimization, and clarify the specific motivation for introducing this component.

2) **[Clarification on Evaluation Protocol in Section 4.3]**
The evaluation in Section 4.3 raises some questions, as the motion generated by Sora may not precisely align with the ground-truth motion. It would be helpful if the authors could explain how alignment between Sora-generated motion and the ground truth is handled when computing the metrics.

3) **[Assumptions Regarding Input Video]**
I am also curious whether there are any assumptions or constraints regarding the input video. Since the proposed method leverages the input to handle occlusions, how does it perform when the occluded regions are not visible in the input video either? If the quality of the results is sensitive to the occlusion coverage in the input, I would suggest providing some analysis or discussion on this aspect.

4) **[Request for Additional Ablations]**
Finally, I believe that additional ablation studies with qualitative comparisons—as mentioned in the Weakness section—would greatly strengthen the paper and help better assess the contribution of each component. Including such analysis would positively impact my overall evaluation.

**Ethical Concerns:**

["NO or VERY MINOR ethics concerns only"]

**Final Justification:**

My primary concern was the novelty over the naive ARAP.
The authors have sufficiently addressed this with ablations and baseline comparisons.
I believe the paper merits acceptance.

**Limitations:**

This paper does not discuss potential societal impacts, which is important given the nature of generative video models.
Additionally, the rationale behind L316–317 in the Limitations section is unclear. Why are highly deformable or textureless objects less temporally consistent? Further explanation would be helpful.

**Paper Formatting Concerns:**

The paper is well-formatted, and I did not observe any formatting issues.

**Quality:**

3

**Strengths And Weaknesses:**

Strengths
1) The paper introduces a new and meaningful task—video-conditioned 4D content generation—effectively bridging generative models with 4D reconstruction.
2) The proposed occlusion-aware rigidity regularization and disocclusion backtracing are thoughtfully designed to address visibility challenges such as occlusion and disocclusion in dynamic scenes.
3) The method demonstrates strong generalizability across diverse object types, including both articulated and rigid objects.

Weakness
1) **[ARAP Regularization and Missing Analysis]**
 ARAP regularization is widely used in 4D reconstruction, so its inclusion alone is not novel. Although the paper introduces an occlusion-aware variant, its impact is unclear without ablation comparing it to standard ARAP. The lack of citations (e.g. SC-GS) and comparison with existing ARAP-based methods further limits the clarity of the contribution.

2) **[Additional ablation on each component]**
Qualitative comparisons showing the effect of each module—e.g., synthetic video optimization only, joint optimization, +ARAP, +occlusion-aware ARAP, and +disocclusion backtracking—would help clarify how each part contributes under different conditions (e.g., occlusion, disocclusion). As it stands, the effectiveness of individual modules difficult to recognize.

3) **[Missing Comparison with Image + Text Methods]**
The paper lacks comparison with existing methods that generate video content conditioned on a single image and a text prompt. Including such a baseline would help contextualize the performance of the proposed approach.

4) **[Unconvincing Visual Results in Figure 6]**
The output in Figure 6, particularly the camel example, is not visually compelling. Artifacts appear between the camel's humps, which makes the result look worse than the driving video itself. It is also unclear which part of the image demonstrates the claimed motion artifact improvements—highlighting or marking these regions would be helpful.

5) **[Missing Detailed Description on Prompt Refinement Using VLM]**
The paper needs justification for why smooth camera motion is necessary. An ablation on VLM refinement is missing, and the instructions used for VLM are not provided, limiting clarity and reproducibility.

6) **[Missing Detailed Description on Experimental Settings]**
The paper needs sufficient details on key experimental settings. For example, it is unclear how the "naive reconstruction" baseline in Figure 4 is implemented. Visualizations of both training and occlusion views would help clarify the effectiveness of the method. In Table 1, the baseline method referenced is not clearly specified—further explanation is needed. Additionally, it is ambiguous whether the ARAP loss is applied to all points or only to occluded regions. Lastly, the number of input video frames and the number of frames in the synthesized video should be explicitly stated for reproducibility.

---

> ### Author Rebuttal · Authors · 2025-07-31
>
> We appreciate that Reviewer `dprx` considers the proposed task 4D Restaging meaningful, the components are thoughtfully designed, and the pipeline generalizable.
>
> ## [Weakness 1] Comparison with existing ARAP methods
> - **The goal of Occlusion-aware ARAP is to better serve a unified pipeline that leverages monocular input videos for controllable deformable scene generation.**
> The design of occlusion-aware ARAP is tailored to the task of 4D restaging, because we aim to transfer structure and articulation from the base video into occluded or disoccluded regions during restaging. Unlike existing methods that rely on multiview or synthetic data, we operate on monocular real-world videos with newly generated motions, where occlusion patterns can differ significantly from the original sequence. By weighting the ARAP loss with visibility cues, we can preserve shape consistency even under occlusion, which is critical for high-quality 4D restaging.
> - **Compare with other methods.**
> We compared our Restage4D with SC-GS and Vidu4D on the DAVIS dataset and reported the PSNR to evaluate reconstruction quality. We ran a sequence of `horsejump-high` and `flamingo`.
> We describe the qualitative and quantitative results here:
> - *Sequence 1*:
>     - *Input video: A rider jumps a horse over an obstacle.*
>     - *Text prompt: "The horse turns around."*
>     - *Reconstruction results:*
>         - *SC-GS fails to model realistic deformation; the horse remains static, with rotating artifacts.*
>         - *Vidu4D reconstructs the foreground reasonably well but lacks background and suffers from blurred edges.*
>         - ***Ours** successfully reconstructs the full scene (foreground and background) with realistic deformation and consistent motion.*
> - *Sequence 2*:
>     - *Input video: A flamingo drinks water with head motion.*
>     - *Text prompt: "The flamingo moves its leg up and down."*
>     - *Reconstruction results:*
>          - *SC-GS fails to animate the leg, which remains static. The entire flamingo even disappears due to camera motion in some frames.*
>          - *Vidu4D only reconstructs the foreground with a gray background. The flamingo follows the prompt.*
>          - ***Ours** produces a full-scene reconstruction with consistent leg motion and high visual quality.*
>     |**Method**   | **horsejump-high-turn** | **flamingo-leg** |
>     | -------- | -------- | -------- |
>     | SC-GS     | 22.41     | 21.57     |
>     | Vidu4D     | 24.2 (foreground)      | 26.4 (foreground)      |
>     | **Ours**     | **25.8**     | **28.91**     |
>
> The results show that existing ARAP-based methods are more suitable for "effective multiview" [1] synthetic data like D-NeRF [2], and fail to reconstruct the motion and geometry for the objects in monocular causal video. In addition, Vidu4D concentrates on reconstructing the foreground, and we only got a background-free object.
> ## [Weakness 2 & Q1 & Q4] Qalitative Ablation
> We provide qualitative description to illustrate the contribution of each module in our pipeline.
> - **Setup.**
> We use the bear sequence from DAVIS with the prompt “The bear walks to the heap.” In the base video, the left side of the bear is visible, while the right side is occluded. In the generated video, the bear turns right and reveals the right-back side of its body (i.e., a disocclusion). This setup naturally introduces both occlusion and disocclusion regions. Then we visualize:
>     - Frame 15 with the camera rotated to the left. It renders the face of the bear, which is occluded now due to its turning right.
>     - Frame 40 with the camera rotated to the right. It renders the right part of the bear, which is disoccluded, revealed by its motion.
> - **Qualitative Analysis.**
> From the visual results, we observe
>   1. **Synthetic video optimization only** will lead to the wrong geometry for the occluded part. The entire face is missing in frame 15. The geometry of the right part in frame 40 is right.
>   2. **1 + ARAP** can help keep a part of the geometry for occlusion. The nose of the bear is missing in frame 15, and the right part in frame 40 is similar to case 1.
>   3. **Joint Optimization** visually it seems not to bring direct benefits, compared to case 1, though, it connects the base video that serves as the chance to transfer the geometry and articulation.
>   4. **3 + ARAP** can utilize the information in the base video, which leads to a good geometry for occlusion in frame 15, keeping the entire head. However, splats are missing for the disoccluded part in frame 40, since it's too rigid with the canonical frame, where these splats are invisible.
>   5.  **3 + Occlusion-aware ARAP** loses the regularization for the dis-occlusion part, leading to a better geometry in frame 40 compared to case 4, and the geometry for occlusion in frame 15 is slightly worse than case 4 for the weaker regularization.
>   6.  **5 + Backtracing** helps to improve the quality of dis-occlusion part and speeds up convergence for its geometry. The splats are denser for the right part in frame 40.
> - **Motivation for Backtracing.**
> Although joint training and occlusion-aware ARAP are still limited in recovering disoccluded regions, which often lack initialization due to not being visible in the base video. Moreover, ARAP tends to over-regularize these areas and prevent new geometry from emerging, especially combined with low-rank motion. Therefore, disocclusion backtracing helps effectively add new points during initialization.
>
> ## [Weakness 3] Other I2V Models
> - **We integrate other image-to-video models into our pipeline, and it works on multiple models.**
> We evaluated CausVid [3] and LTX-Video [4] for video generation and compared them with Sora. We applied our restaging pipeline to two sequences: cat-jump and cows-graze, using each I2V model as the video generation component. Below are the PSNR scores of the final restaged outputs:
>     | I2V Model | cat-jump | cows-graze |
>     | -------- | -------- | -------- |
>     | Sora     | 26.74     | 26.25     |
>     | CausVid     | 23.68    | 26.45     |
>     | LTX-video     | 25.08     | 28.05     |
>
>     These results demonstrate that our Restage4D framework performs effectively across diverse I2V backbones, with consistent reconstruction quality and visual plausibility.
> ## [Weakness 4] Result in Figure 6
> It's difficult to show the dynamic results using only images. We provide the video illustration for Figure 6 in the supplementary (3:53-4:10).
> ## [Weakness 5 & Q2] VLM Refinement & Evaluation
> - **Criteria to pick the video**
> Currently, the generated driving videos are manually filtered to ensure basic quality:
> 	1.	Camera: We discard sequences with hard cuts or abrupt viewpoint changes, as these cause failures in dense point tracking and violate the motion consistency assumption.
> 	2.	Motion: We also exclude sequences where the motion does not align with the input text prompt, in order to ensure edit fidelity.
> - **Effect of smooth camera**
> The smoothness of the camera is essential for motion initialization. We use the tracking point from Tapir [5] to initialize the foreground Gaussian Splats, so we evaluate Tapir on sequences generated conditionally on `car-roundabout` in DAVIS with the prompt "The car drives away". One of the sequences has smooth camera motion and another has hard cuts. Tapir performs well on the smooth sequence, but on the hard-cut one, it fails to find the correspondence before and after the cut, leading to some floating points.
> - **Ablation on VLM Refinement**
> Using VLM to refine the prompt can provide a more precise description of the camera and motion control, which accelerates the filtering for videos. In the `car-roundabout` case, with VLM refinement, there are 5 videos out of 8 that satisfy the criteria using Sora, and only 1 out of 8 without refinement. We will provide the prompt for refinement in our final version.
> ## [Weakness 6] Experimental Settings
> - **Figure 4 & Table 1**
> Naïve reconstruction and baseline refer to Shape-Of-Motion.
> - **Range of ARAP**
> During the motion initialization stage, ARAP is applied to all points for better rigid motion and geometry initialization, while during the photometric fusion stage, ARAP is applied to occlusion points.
> - **Training Details**
> The base video has 50 to 60 frames, and the driving video has 40 to 50 frames. We will provide the precise number.
> ## [Q3] Assumptions Regarding Input Video
> Since we utilize the input video to learn the geometry and articulation, the pipeline works best when the motion and geometry in the input video are richer than those in the generated video. If the occluded regions are also not visible in the input video, the model can't reconstruct the geometry since we don't incorporate a generative prior during reconstruction.
> ## [Limitation]
> - **Social Impact**
> Our framework involves generative video models, which may pose risks when applied to human-centric content, potentially leading to the synthesis of misleading or harmful videos. One possible solution is incorporating prompt filtering mechanisms during the prompt refinement stage to detect and block inappropriate or sensitive prompts.
> - **Performance Limitation**
> Our method relies on existing dense point tracking for motion initialization. However, for highly deformable or textureless objects, such tracking often fails due to ambiguous correspondences across frames. This leads to temporal inconsistencies in the reconstructed geometry and motion. Improving robustness under such cases remains a promising direction for future work.
>
> We will add these discussions and show all the figures in the revised version.
>
> [1] Monocular dynamic view synthesis: A reality check.
>
> [2] D-nerf: Neural radiance fields for dynamic scenes.
>
> [3] From slow bidirectional to fast autoregressive video diffusion models.
>
> [4] Ltx-video: Realtime video latent diffusion.
>
> [5] Tapir: Tracking any point with per-frame initialization and temporal refinement.

---

> > ### Author Response · Authors · 2025-08-02
> >
> > Dear reviewer `dprx`,
> >
> > Please let us know if we solved your concerns. We are happy to discuss if you have any questions. Your feedback means a lot to us.

---

> > > ### Comment · Reviewer_dprx · 2025-08-03
> > > **ARAP ablation study**
> > >
> > > Could you provide the quantitative ablation results related to ARAP?
> > > I would like to see a numerical comparison between the ARAP (not occlusion-aware) and the occlusion-aware ARAP in the ablation study.
> > > If I have misunderstood anything, please feel free to correct me.
> > > I tried to find these ablation results, but I wasn’t able to find them.

---

> > > > ### Author Response · Authors · 2025-08-03
> > > > **ARAP Ablation Study**
> > > >
> > > > Thank you for your insightful question. We conducted quantitative ablation experiments comparing the standard (naive) ARAP with our proposed occlusion-aware ARAP, using the same settings as in Table 1. The results are shown below:
> > > >
> > > >
> > > > | Method | PSNR $\uparrow$ | OCV CLIP $\uparrow$ | Vol. Consistent $\uparrow$| Edge Consistent $\uparrow$ |
> > > > | -------- | -------- | -------- | -------- | -------- |
> > > > | Baseline     | $26.71\pm1.97$  | $81.68\pm4.91$  |$7.41\pm3.80$ | $12.66\pm3.01$ |
> > > > | +Joint + Naive ARAP | $21.47\pm1.94$  | $87.31\pm3.05$  |$11.43\pm4.27$ | $17.30\pm3.95$ |
> > > > | $\Delta$ from Baseline | **-5.24**  | $+5.63$  |$+4.02$ | $+4.64$ |
> > > > | +Joint + Occlusion-aware ARAP     | $27.12\pm2.27$  | $85.40\pm4.11$  |$10.32\pm4.23$ | $16.28\pm3.10$ |
> > > > | $\Delta$ from Baseline | $+0.41$  | $+3.72$  |$+2.91$ | $+3.62$ |
> > > >
> > > > These results reflect our core motivation for designing the occlusion-aware ARAP. While the naive ARAP improves geometric consistency metrics (Vol./Edge Consistency) for the occluded part, it applied a strong rigidity constraint globally, which significantly degrades the reconstruction quality (PSNR drops by 5.24). This is especially problematic in regions with high-frequency motion, where excessive regularization results in visually unnatural and frozen movements, leading to incorrect motions.
> > > >
> > > > Through empirical observations, we found a trade-off:
> > > >
> > > > - Enforcing ARAP uniformly helps preserve occluded geometry, but overly constrains visible regions and suppresses natural motion.
> > > > - Reducing regularization improves visible motion, but sacrifices occluded region stability.
> > > >
> > > > This tension made us realize that the level of regularization should be different for visible and invisible parts, and led us to propose the occlusion-aware ARAP, which applies stronger regularization only to occluded regions and relaxes it elsewhere. As shown, it achieves a better overall balance: strong consistency while maintaining high visual quality (PSNR increased by 0.41 over baseline).
> > > >
> > > > We will state this more clearly in our revised version. Thank you again for your question.

---

> > > > > ### Comment · Reviewer_dprx · 2025-08-04
> > > > >
> > > > > Thank you for the quick and thorough response, along with the additional ablation results.
> > > > > I have a few follow-up questions:
> > > > >
> > > > > 1. Could you also provide the ablation results for ARAP alone (without Joint)?
> > > > > In Table 1, I noticed that there is a variant with only +ARAP (w/o Joint).
> > > > > For a clearer understanding of the effect of occlusion-aware ARAP, I believe comparing it against this ARAP-only variant (without joint supervision) would be more straightforward.
> > > > > Could you include the corresponding results using the naive ARAP (non-occlusion-aware) under the same setting?
> > > > >
> > > > > 2. It would be helpful if you could also report additional visual quality metrics (e.g., SSIM, LPIPS) alongside PSNR.
> > > > > Demonstrating consistent improvements across these metrics would strengthen your claim.
> > > > > Currently, since the occlusion-aware ARAP only improves PSNR but falls short on all other reported metrics, it's difficult to confidently conclude that it is overall better.
> > > > >
> > > > > Thank you again for your thoughtful responses and for considering these additional points.

---

> > > > > > ### Author Response · Authors · 2025-08-05
> > > > > >
> > > > > > Thank you for your thoughtful suggestions. We conducted the requested ablation comparing naive ARAP and occlusion-aware ARAP, both applied without joint optimization (i.e., using only the driving video), under the same settings. The results are shown below:
> > > > > > | Method | PSNR $\uparrow$ | SSIM $\uparrow$ | LPIPS $\downarrow$ | OCV CLIP $\uparrow$ | Vol. Consistent $\uparrow$| Edge Consistent $\uparrow$ |
> > > > > > | -------- | -------- | -------- | -------- | -------- | -------- | -------- |
> > > > > > | Baseline     | $26.71\pm1.97$ | $0.9756\pm 0.004$ | $0.0994\pm 0.049$| $81.68\pm4.91$  |$7.41\pm3.80$ | $12.66\pm3.01$ |
> > > > > > | + Naive ARAP | $23.67\pm1.78$ | $0.9581\pm 0.007$ | $0.1733\pm 0.091$| $84.78\pm4.39$  | $9.14\pm3.31$  |$15.91\pm3.84$ | $17.30\pm3.95$ |
> > > > > > | $\Delta$ from Baseline | **-3.04** | **-0.0175**  | **+0.0739**  | $+3.1$  |$+1.73$ | $+3.25$ |
> > > > > > | + Occlusion-aware ARAP     |$26.80\pm2.26$ |$0.9812\pm 0.006$ |  $0.0754\pm0.044$ | $82.65\pm4.29$  |$7.92\pm3.97$ | $13.64\pm3.38$ |
> > > > > > | $\Delta$ from Baseline | $+0.19$  | $+0.0056$  |$-0.024$ | $+0.97$ | $+0.51$ | $+0.98$
> > > > > >
> > > > > > These results show a consistent trend with the joint optimization setting:
> > > > > > - While naive ARAP improves geometric consistency metrics (Vol./Edge Consistency and OCV CLIP), it significantly degrades visual quality, with PSNR dropping by 3.04, SSIM decreasing, and LPIPS increasing, which indicates over-regularized motion.
> > > > > > - In contrast, occlusion-aware ARAP achieves strong visual quality across all three metrics (PSNR, SSIM, LPIPS) while still improving geometric consistency relative to the baseline.
> > > > > >
> > > > > > This validates our design motivation: uniformly enforcing ARAP over both visible and occluded regions imposes overly strong regularization on high-frequency visible motion, leading to artifacts and frozen appearances. By selectively regularizing only the occluded regions, occlusion-aware ARAP achieves a better balance between geometry preservation and visual quality.
> > > > > >
> > > > > >
> > > > > > We will include this discussion and additional results in the revised version. We truly appreciate your detailed and constructive feedback, and hope this can address your concerns.

---

> > > > > > > ### Comment · Reviewer_dprx · 2025-08-05
> > > > > > >
> > > > > > > I appreciate the authors' response.
> > > > > > >  My concerns have been fully addressed, and I will raise my scores.

---

### Note · Authors · 2025-08-15

Dear Area Chairs,

We sincerely thank all reviewers for their thoughtful reviews and constructive feedback. During the rebuttal phase, we engaged in active discussions and addressed the major concerns raised, both through additional experiments and clearer explanations of our design choices.

Reviewer `dprx` appreciated the motivation behind our proposed task of 4D Restaging, and acknowledged that our pipeline is well-structured and generalizable. Their main concern centered around the need for more comprehensive ablations, particularly regarding the occlusion-aware ARAP. In response, we provided detailed quantitative and qualitative comparisons with its naive counterpart, as well as additional perceptual metrics (SSIM, LPIPS). These results validated our motivation and demonstrated that occlusion-aware ARAP achieves a better trade-off between geometric consistency and visual fidelity. The reviewer acknowledged this clarification and expressed satisfaction with the response.

Reviewer `gQQs` found the task interesting and novel, and agreed with our problem formulation. Their concerns focused on the complexity of the task design, the evaluation protocol, and the significance of the problem setting. We responded by clarifying the motivation behind our video-driven approach, its benefits over existing paradigms, and how our components (e.g., frame selection and motion supervision) support the pipeline holistically. While we acknowledged some differences in problem framing, the reviewer appreciated the design of our occlusion-aware rigidity enforcement and recognized the value of our contributions.

Reviewer `duUh` viewed the task as engaging and the pipeline as insightful. To address their request, we provided additional results on complex scenes involving multiple interacting objects, which helped demonstrate the scalability of our method. The reviewer maintained a positive view after the rebuttal.

Reviewer `Nx6n` found our results promising and requested comparisons with additional baseline methods on both DAVIS and Neural3DV. We conducted these experiments across a larger set of sequences and showed that our method consistently outperforms baselines across datasets. This successfully addressed the reviewer’s concerns.

Overall, we believe the rebuttal process helped significantly strengthen the paper. We appreciate the opportunity to clarify our work and hope the final decision reflects the technical merit and novelty of our contributions.

---

### Decision · Program_Chairs · 2025-09-17

**Decision:**

Accept (poster)

**Comment:**

This paper proposes a new task of generating 4D content by conditioning on a single real-world video.
It receives two acceptance and two boardline acceptance.
Main concerns are about ablation studies, state-of-the-art comparisons.
The authors provided extensive experiments to resolve the concerns.
The AC recommends acceptance for this paper.
However, the AC found the abstract in the openreview is not consistent with the version in the submitted pdf file. The authors are urged to update and revise any inconsistencies.